# Non-Invasive Myocardial Work in Patients with Severe Aortic Stenosis

**DOI:** 10.3390/jcm11030747

**Published:** 2022-01-29

**Authors:** Salvatore De Rosa, Jolanda Sabatino, Antonio Strangio, Isabella Leo, Letizia Rosa Romano, Carmen Anna Spaccarotella, Annalisa Mongiardo, Alberto Polimeni, Sabato Sorrentino, Ciro Indolfi

**Affiliations:** 1Department of Medical and Surgical Sciences, Magna Graecia University, 88100 Catanzaro, Italy; jolesbt@hotmail.it (J.S.); antonio91strangio@gmail.com (A.S.); isabella.leo98@gmail.com (I.L.); leti.romano7@gmail.com (L.R.R.); spaccarotella@unicz.it (C.A.S.); mongiardo@unicz.it (A.M.); polimeni@unicz.it (A.P.); sorrentino@unicz.it (S.S.); 2Mediterranea Cardiocentro, 80122 Naples, Italy

**Keywords:** myocardial work, aortic stenosis, speckle tracking analysis

## Abstract

Changes in cardiac mechanics after correction of severe Aortic Stenosis (AS) are under-investigated. Myocardial Work (MW) is emerging as a useful non-invasive correlate of invasively measured myocardial performance and oxygen consumption. The aim of this study was to assess the usefulness of MW in the clinical management of patients with AS undergoing transcatheter aortic valve implantation (TAVI). Consecutive patients referred for TAVI were included in this observational study. Echocardiograms were performed before and after TAVI to measure Global Work Index (GWI), Global Constructive Work (GCW), Global Wasted Work (GWW), and Global Work Efficiency (GWE). Mean transvalvular gradient was significantly improved (*p* < 0.05), without significant changes in left ventricular ejection fraction, nor in global longitudinal strain (GLS). GWI (*p* < 0.001) and GCW (*p* < 0.001), but not GWW (*p* = 0.241) nor GWE (*p* = 0.854) were significantly reduced after TAVI. Patients with a low flow low gradient (LF-LG) AS had lower left ventricular ejection fraction (LVEF) (*p* < 0.001), worse global longitudinal strain (GLS) (*p* < 0.001), and lower baseline GWI (*p* < 0.001), GCW (*p* < 0.001) and GWE (*p* = 0.003). The improvement in GWI and GCW observed after TAVI in the general study population were abolished among LF-LG patients. In conclusion, non-invasive MW might be useful to further classify patients with AS and to predict non responders.

## 1. Introduction

Severe aortic stenosis (AS) is associated with variable degrees of compensatory left ventricular (LV) adaptation [1]. These changes maintain the stroke volume despite the presence of a mechanical obstacle, but they subtly lead to LV dysfunction, and consequently to a rapid clinical progression and a steep worsening of prognosis [2].

Assessment of systolic LV performance in these patients represents a clinical conundrum. Left ventricular ejection fraction (LVEF) generally remains preserved despite initial deterioration of LV function. Global longitudinal strain (GLS) has been shown to be an earlier and more sensitive marker of LV dysfunction [3,4]. Nevertheless, its dependence from afterload represents a limitation in severe AS [5].

Noninvasive myocardial work, a non-invasive correlate of myocardial performance and oxygen consumption, has been introduced, it uses segmental strain and non-invasive blood pressure to calculate LV pressure–strain loops [6,7,8]. 

In this context, the aim of the present study was to assess the feasibility of non-invasive myocardial work assessment in patients with severe AS, the impact of stenosis correction by means of transcatheter aortic valve implantation (TAVI), and to investigate its usefulness for pre-procedural assessment.

## 2. Materials and Methods

### 2.1. Study Population

In this single-centre study, consecutive patients with symptomatic severe AS scheduled for transcatheter aortic valve implantation (TAVI) at the Cardiology Department of the University Magna Graecia of Catanzaro (Italy) from January 2018 to August 2019 were included. Severe AS was defined by an aortic valve area below 1 cm^2^ or an aortic valve index ≤ 0.6 cm^2^ per square meter. Low flow low gradient (LF-LG) severe aortic stenosis is defined as an aortic valve AVA ≤ 1.0 cm^2^ or indexed ≤ 0.6 cm^2^/m^2^ and a mean transvalvular gradient < 40 mmHg. Surgical risk was assessed using the Euroscore II. Clinical management was guided through collegial discussion of the Heart Team, according to current guidelines. Exclusion criteria were: hemodynamically relevant aortic regurgitation (more than mild), presence of left bundle branch block at EKG, suboptimal quality of the echocardiographic images. The regional ethics committee approved the study, and all the patients provided written informed consent.

### 2.2. Study Timeline, Procedures, and Analysis Plan

Clinical examination, anamnesis, a 12-lead ECG, and laboratory tests were performed at baseline. Echocardiography was undertaken before and after TAVI for all patients. Non-Invasive Blood Pressure (NIBP) was registered at every examination. For a subgroup of patients, Invasive Blood Pressure (IBP) was measured during TAVI. All clinical data were anonymized and collected in an electronic database.

### 2.3. Echocardiographic Analyses

All patients underwent a standard Trans-Thoracic Echocardiography (TTE) using a Vivid E9 ultrasound system (General Electric Healthcare, Horten, Norway). For the quantification of the severity of aortic stenosis, we measured the left ventricular outflow tract (LVOT) diameter, LVOT flow by pulsed wave doppler (PW), AS jet by continuous wave doppler (CW), and the aortic valve area (AVA) calculated by continuity equation, as recommended. For the strain analysis, two-dimensional (2D) 4-chamber, 3-chamber, and 2-chambers apical views were acquired, with an optimized frame rate (≥60 frames/sec). All the recorded images were transferred to an imaging workstation (Echo PAC version 112.99, Research Release, GE Healthcare), which allowed offline semi-automated analysis.

### 2.4. Calculation of Non-Invasive Myocardial Work

The left ventricular (LV) Myocardial work (MW) is estimated as the area of the LV pressure–strain loop by a dedicated software (GE-Healthcare), as previously described [9]. However, the standardized pressure curve obtained experimentally to calculate MW by Russell et al. was not validated in patients with aortic stenosis, in which higher pressure gradients are observed between the LV and the aorta. For the calculation of Myocardial Work indices, the estimated LV pressure was obtained by adding the mean transvalvular pressure gradient to brachial artery cuff pressure, as previously described [10,11]. Timing of opening and closure of the valve were manually selected from 2D echo images. Global myocardial Work Index (GWI), Global myocardial Constructive Work (GCW), Global myocardial Wasted Work (GWW), and Global myocardial Work Efficiency (GWE) were calculated as previously described [4].

### 2.5. Data Analysis and Statistics

Continuous variables are presented as mean ± standard deviation, categorical as percentages. The Kolmogorov–Smirnov test was used to assess normality. Analysis of Variance (ANOVA) or Student T-Test was used for comparisons with continuous variables. The Chi squared test was used for dichotomous variables. Pearson’s R test was used to assess correlation between variables. Multiple logistic regression was used to test predictors of clinical adverse events (death, HF-related hospitalization), as previously described [12]. Covariates were tested in the multivariable regression model if they were positive in univariable analysis. Bootstrapping with 1000 re-samples was used to confirm results of multivariate regression analysis. A two-tailed *p*-value of 0.05 was considered significant. Corrections for multiple comparisons were applied to statistical analyses. Specifically, as GWI, GCW, GWW, and GWE present are not completely independent, adjustment with the Holm method was applied in this case. To account for comparison in multiple independent subgroups (gender groups, LF-LG), adjustment with the Bonferroni method was applied [13,14]. As myocardial work indices were repeatedly measured in study patients, we applied a repeated measure fixed-effect model to account for time-correlation over time. Using this model, myocardial work indices were used as the within-subject factor, and 4 time-points were considered (pre-TAVI, 2 weeks, 4 months, 1 year) [15].

Based on the previous literature, we estimated a 15% reduction in mean Global Work Index after TAVI in the study population. Thus, we calculated that 70 patients would have been required to detect the estimated change in Global Work Index after TAVI with a 90% power and an alpha level of 0.05.

The statistical analyses were performed using SPSS v.21 (SPSS Inc., Chicago, IL, USA).

## 3. Results

### 3.1. Study Population

During the enrolment period, 73 consecutive patients with symptomatic severe AS scheduled for TAVI fulfilled the inclusion and exclusion criteria and were included. As shown in Table 1, depicting baseline demographics, the study population is composed of elderly patients, with equal gender representation, affected by degenerative valvular aortic stenosis, as confirmed by a mean transvalvular gradient of 46.7 ± 15 mmHg. Cardiac remodeling was present, as shown by the increase in LV hypertrophy, and the pathologic enlargement in left atrial volume. TAVI procedures were performed under monitored sedation, via a transfemoral approach. Three fourth of implanted prostheses were self-expandable.

### 3.2. Estimation of LV Pressure

To assess the feasibility of LV pressure estimation using the mean transvalvular pressure gradient to correct brachial artery cuff pressure, invasive LV pressure measurements were obtained in a subset of 23 patients. In this subgroup, estimated LV pressure compared well to invasively measured LV pressure (r = 0.89; *p* < 0.001). Similarly, the Bland–Altman plot showed good reproducibility (Appendix A).

### 3.3. Assessment of LV Function and Mechanics

Standard baseline echocardiographic parameters are reported in Appendix A. After TAVI, we observed a significant improvement of mean transaortic gradient (<0.001), while no changes in left ventricular ejection fraction were observed (*p* = 0.967). A modest, non-significant change of E/E’ was observed from 13.2 ± 3.0 to 12.7 ± 2.1 after TAVI (*p* = 0.205). Global longitudinal strain after TAVI (−18.8 ± 4.0) was not significantly improved compared to pre-procedural values (−16.8 ± 5.0; *p* = 0.060) (Figure 1).

GWI and GCW were significantly reduced after TAVI and this difference was maintained during the follow-up (Figure 2). On the contrary, we observed no significant change in GWW nor in GWE (Figure 2). These results were maintained after adjustment for repeated measures.

Despite no difference in mean LVEF (*p* = 0.417) nor in GLS (*p* = 0.142) (Appendix A), women had higher GWI (*p* = 0.007) and GCW (*p* = 0.014) compared to men (Figure 3), with no difference in GWW (*p* = 0.763) nor in GWE (*p* = 0.393) at baseline. The delta change of GCW after TAVI was larger in women, although it remained significant in both sex groups (*p* = 0.001 in men; *p* < 0.001 in women). A similar effect was found for GCW (*p* < 0.001 in both groups), while a significant delta change in GWE was measured in women (*p* = 0.014), but not in men (*p* = 0.656). No significant delta change was observed in either group for GWW (Figure 4). These results were maintained after adjustment for multiple comparison.

Patients with a low flow low gradient (LF-LG) severe aortic stenosis presented a lower LVEF (*p* < 0.001) and a worse GLS (*p* < 0.001) compared to high gradient (HG) AS (Appendix A). These patients had significant lower baseline GWI (*p* < 0.001), GCW (*p* < 0.001) and GWE (*p* = 0.003) and no difference in GWW (*p* = 0.605) (Figure 5). The reductions in GWI and GCW observed after TAVI in the study population were abolished in the subgroup of LF-LG patients (*p* = 0.465 for GWI; *p* = 0.465 for GCW) (Figure 6). These results were maintained after adjustment for multiple comparison.

### 3.4. Predictors of Clinical Events

Patients were followed up for a total of 128 patient years at risk. During a median follow-up of 24 months, 11 patients (15%) died and 23 (32%) were re-hospitalized. Of these re-admissions, 7 (10%) were related to HF. Univariate predictors of cardiovascular death included endocarditis (r = 0.285; *p* = 0.018), cigarette smoke (r = 0.326; *p* = 0.005), chronic obstructive pulmonary disease (COPD) (r = 0.281; *p* = 0.017), or presence of mitral regurgitation (r = 0.285; *p* = 0.015). Predictors of HF-related rehospitalization included diabetes (r = 0.268; *p* = 0.027), hypercholesterolemia (r = −0.244; *p* = 0.045) and residual aortic regurgitation after TAVI (r = 0.308; *p* = 0.040), along with baseline GWI (r = −0.359; *p* = 0.003), baseline GCW (r = −0.324; *p* = 0.007), baseline GWE (r = −0.322; *p* = 0.008), delta change in GCW (r = −0.281; *p* = 0.046), or GWE (r = −0.368; *p* = 0.05).

At multivariate analysis, baseline GWI remained an independent predictor of death (B = 0.002; *p* = 0.011), along with powerful predictors such as hypercholesterolemia (*p* = 0.023), mitral regurgitation (*p* = 0.020), and COPD (*p* = 0.019). In addition, GWI was the only independent predictor of HF-related re-hospitalizations (*p* = 0.024).

Similarly, multivariate analysis showed that baseline GCW was an independent predictor of death (B = 0.002; *p* = 0.016), together with hypercholesterolemia (*p* = 0.028), mitral regurgitation (*p* = 0.021), and COPD (*p* = 0.021). In addition, GWI was an independent predictor of HF-related re-hospitalizations (*p* = 0.050) together with hypercholesterolemia (*p* = 0.046).

Neither baseline GWE, nor GWW were independent predictors of death or HF-related re-hospitalization.

## 4. Discussion

In the present study, we assessed the impact of mechanical correction of severe valvular AS by means of TAVI on non-invasive myocardial work parameters. The main results of our study are that (i) correction of severe AS by means of TAVI has a significant impact on non-invasive myocardial work parameters, which are maintained over time during the follow-up; (ii) non-invasive myocardial work indices are capable to further stratify patients with severe AS, as they were able to discriminate between LF-LG AO and the more common form of AS with HG; (iii) the improvement of myocardial work indices after TAVI is not equally present across patient subgroups, as women tended to experience a less pronounced delta change in myocardial work after TAVI. The delta change is apparently abolished in specific subgroups, such as in patients presenting with LF-LG severe AS; (iv) myocardial work indices are independent prognostic predictors of adverse clinical outcomes.

The evidence that baseline myocardial work indices are able to independently predict adverse clinical events pairs well with recent results reporting a significant correlation between non-invasive myocardial work and clinical symptoms in AS patients [11]. In addition, the finding that LF-LG patients have worse myocardial work indices at baseline pairs with the lack of improvement of the same parameters after TAVI, suggesting that particular caution should be used when selecting candidates to TAVI among patients with LF-LG severe AS.

Our results further confirm previous findings by other groups, that correction of peripheral blood pressure by the transvalvular pressure gradient to estimate LV pressure in patients with AS is feasible and effective. In fact, Jain et al. recently assessed LV mechanics after TAVI in 35 patients with severe aortic stenosis. They found a significant improvement in GLS with a parallel reduction in GWI [10]. Since GWI and the other myocardial work parameters were validated using non-invasive, peripheral blood pressure, the presence of hemodynamically relevant aortic stenosis poses a critical issue to the application of this analysis in AS patients. Hence, they applied a correction of peripheral blood pressure values using the transvalvular pressure gradient and found that it was reliable, as they reported a close correlation between the left ventricular systolic pressure (LVSP) measured invasively and the estimated LVSP obtained by simple addition of mean aortic gradient by means of echocardiography to the peripheral cuff pressure [10]. In this regard, our data further confirm their results, as we similarly found a very close correlation between invasively measured LVSP and estimated LVSP obtained using the same method. In line with their results, we also found a reduction in GWI and GCW after TAVI. A similar protocol was also adopted by Fortuni F. and colleagues [11], who described how myocardial work indices correlate with HF symptoms in patients with severe AS. In line with those observations, our findings showed that patients with LF-LG severe AS, who present a significantly worse LV function in terms of LV ejection fraction and GLS, present worse baseline myocardial work indices. In addition, these latter patients stand out as those not manifesting any improvement in myocardial work indices after TAVI. Finally, our finding that worse myocardial work indices are independent prognostic predictors further extends results by Fortuni and colleagues, suggesting that these indices might be useful prognostic markers for risk stratification and selection of TAVI candidates. This latter hypothesis demands further clinical investigation in larger multicenter studies.

## Figures and Tables

**Figure 1 jcm-11-00747-f001:**
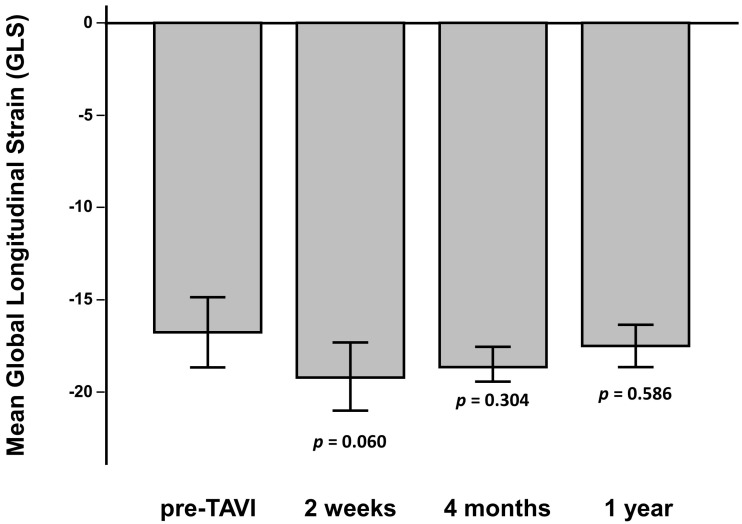
Mean Global Longitudinal Strain (GLS) at multiple study time points. *p* values reported on each bar refer to comparisons with pre-TAVI values.

**Figure 2 jcm-11-00747-f002:**
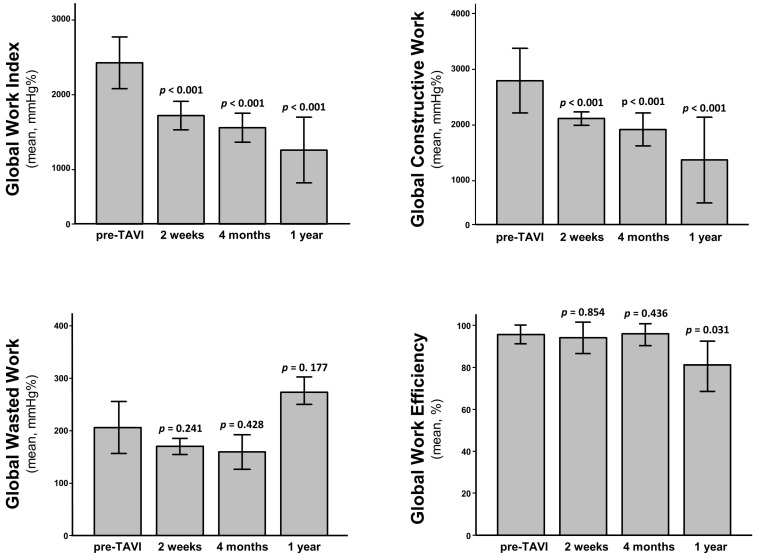
Mean MW indices at multiple study time points: Global Work Index (**upper left**), Global Constructive Work (**upper right**), Global Wasted Work (**lower left**), and Global Work Efficiency (**lower right**). *p* values reported on each bar refer to comparisons with pre-TAVI values. Not statistically significant after adjustment for repeated measures and multiple comparison.

**Figure 3 jcm-11-00747-f003:**
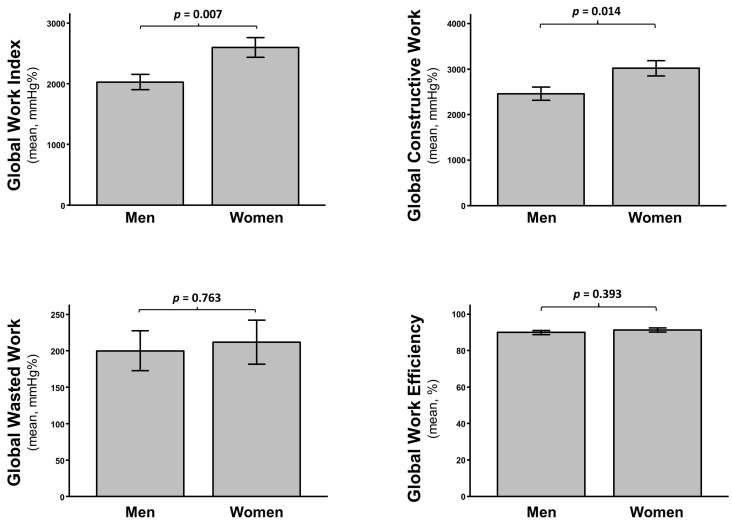
Mean baseline MW indices in gender subgroups: Global Work Index (**upper left**), Global Constructive Work (**upper right**), Global Wasted Work (**lower left**), and Global Work Efficiency (**lower right**).

**Figure 4 jcm-11-00747-f004:**
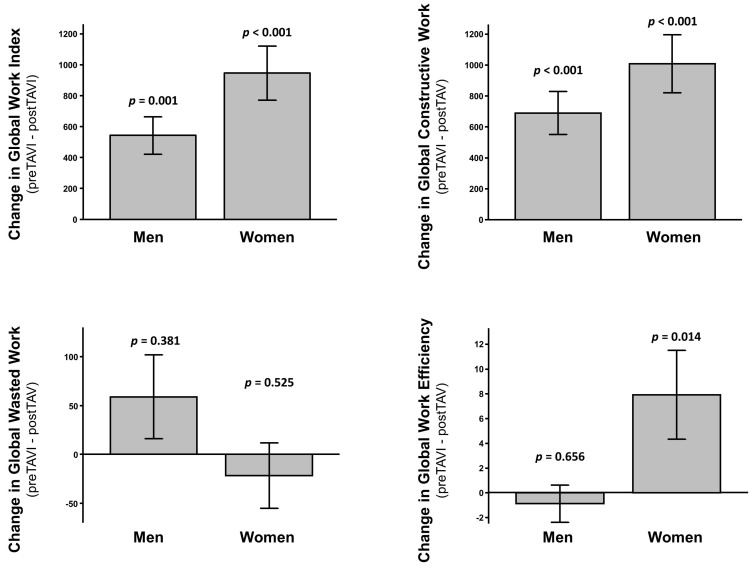
Delta change in MW indices according to sex groups: Global Work Index (**upper left**), Global Constructive Work (**upper right**), Global Wasted Work (**lower left**), and Global Work Efficiency (**lower right**).

**Figure 5 jcm-11-00747-f005:**
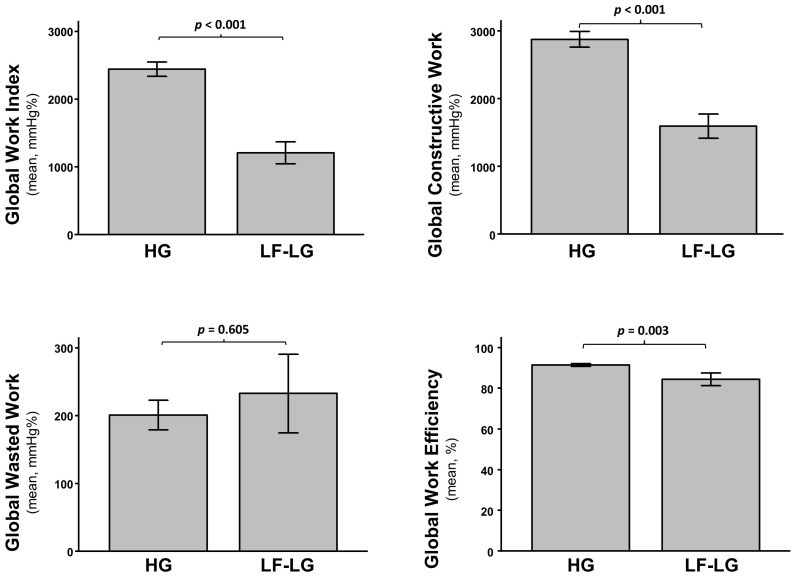
Comparison of mean baseline MW indices between patients with normal high gradient severe aortic stenosis (HG) and low flow/low gradient severe aortic stenosis (LF-LG): Global Work Index (**upper left**), Global Constructive Work (**upper right**), Global Wasted Work (**lower left**), and Global Work Efficiency (**lower right**).

**Figure 6 jcm-11-00747-f006:**
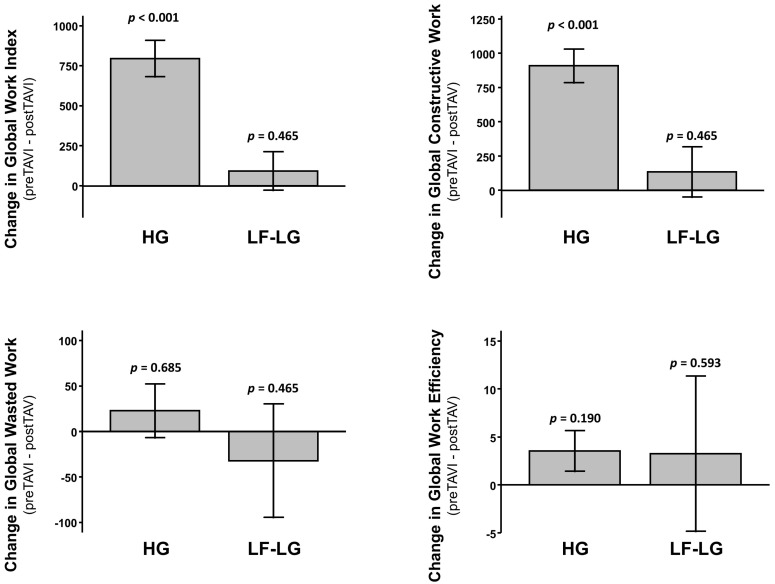
Delta change in MW indices between patients with normal high gradient severe aortic stenosis (HG) and low flow/low gradient severe aortic stenosis (LF-LG): Global Work Index (**upper left**), Global Constructive Work (**upper right**), Global Wasted Work (**lower left**), and Global Work Efficiency (**lower right**).

**Table 1 jcm-11-00747-t001:** Baseline Patient Characteristics.

	(*n* = 73)
Age (years)	80.8 ± 6
Female, *n* (%)	34 (46.5)
Weight (kg)	69.9 ± 12.7
Height (cm)	160.6 ± 7.6%
BSA (m^2^)	1.76 ± 0.18
BMI (kg/m^2^)	26.9 ± 4.19
AVA (cm^2^)	0.78 ± 0.16
AV mean gradient (mmHg)	46.7 ± 15.8
AV peak gradient (mmHg)	74 ± 22.1
LVEF (%)	52.8 ± 9.3
LVEDD (mm)	50.6 ± 7.1
LVESD (mm)	34.5 ± 8.5
IVSd (mm)	14.1 ± 3
LAVi (mL/m^2^)	43.8 ± 8.4
TAPSE (mm)	21.5 ± 3.1
PASP (mmHg)	41.6 ± 10.9
GLS (%)	−16.8 ± 5
Low-Flow Low-Gradient, *n* (%)	10 (13.7)
Hypertension	69 (94.5)
Diabetes Mellitus	17 (23.3)
Dyslipidemia	45 (61.6)
Prior PCI, *n* (%)	14 (19.2)
Self-Expandable Valve, *n* (%)	56 (76.7)
Ballon-Expandable Valve, *n* (%)	17 (23.3)
Valve Dimension (mm)	
23, *n* (%)	17 (23.3)
25, *n* (%)	1 (1.4)
26, *n* (%)	24 (32.9)
29, *n* (%)	23 (31.5)
34, *n* (%)	8 (10.9)

BSA, Body Surface Area; BMI, Body Mass Index; AVA, Aortic Valve Area; AV, Aortic Valve; LVEF, Left Ventricular Ejection Fraction; LVEDD, Left Ventricular End-Diastolic Diameter; LVESD, Left Ventricular End-Systolic Diameter; IVSd, Interventricular Septal Diameter diastolic; LAVi, Left Atrial Volume index; TAPSE, Tricuspid Annular Plane Excursion; PASP, Pulmonary Artery Systolic Pressure; GLS, Global Longitudinal Strain; PCI, percutaneous coronary intervention.

## Data Availability

Deidentified study data are available upon request from Salvatore De Rosa (saderosa@unicz.it).

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
