# Peer review of "Non-Invasive Myocardial Work in Patients with Severe Aortic Stenosis"

_jcm, 2022, doi:10.3390/jcm11030747_

Round 1
Reviewer 1 Report
Thank you for allowing me to review the following manuscript titled "Non-invasive myocardial work in patients with severe aortic 2 stenosis" by De Rosa et al
Abstract: GWI GCW GWW GWE are not expanded
define low flow low gradient
what is the exclusion criteria It is not clear
lone 198 in discussion change " LG-LG AO' to LF LG
Author Response
Thank you for allowing me to review the following manuscript titled "Non-invasive myocardial work in patients with severe aortic 2 stenosis" by De Rosa et al
Authors’ response: we thank the Reviewer for taking the time to read and revise our manuscript. We addressed all comments and suggestion, as reported below.
Abstract: GWI GCW GWW GWE are not expanded
Authors’ response: we thank the Reviewer to highlight this incongruence. We have now revised the abstract, spelling out the above cited acronyms.
define low flow low gradient
Authors’ response: we thank the reviewer for this suggestion. We added the definition adopted for low flow low gradient aortic stenosis, which is widely used in literature, in the methods section of the revised manuscript, as reported below:
“Low flow low gradient (LF-LG) severe aortic stenosis is defined as an aortic valve AVA ≤ 1.0 cm2 or indexed ≤ 0.6 cm2/m2 and a mean transvalvular gradient < 40 mmHg.”
what is the exclusion criteria It is not clear
Authors’ response: we thank the Reviewer for this question. The description of exclusion criteria has been now enlarged in the revised manuscript, as reported below:
“Exclusion criteria were: hemodynamically relevant aortic regurgitation (more than mild), presence of left bundle branch block at EKG, suboptimal quality of the echocardiographic images.”
lone 198 in discussion change " LG-LG AO' to LF LG
Authors’ response: we apologize for the typing error. We revised the sentence in the discussion as suggested.
Reviewer 2 Report
Salvatore De Rosa et. al provide a well written manuscript on non-invasive myocardial work in patients with AS and for patients treated using TAVI. It is a single center study including 70 patients. The authors provide a detailed overview on a variety of non-invasive myocardial work parameter assessed using echocardiography.
I have some comments mainly referred to the methodology which will hopefully further improve the quality of the manuscript.
- Can you please provide further information regarding the variable selection of your multivariable regression model.
- Can you provide some information regarding the fact that you used an alpha of 0.005 in your sample size estimation.
- But the most important issue in my eyes is the great amount of tested hypotheses. You should definitely implement adjustment for multiplicity in your analysis, in order to draw firm conclusions.
Author Response
Salvatore De Rosa et. al provide a well written manuscript on non-invasive myocardial work in patients with AS and for patients treated using TAVI. It is a single center study including 70 patients. The authors provide a detailed overview on a variety of non-invasive myocardial work parameter assessed using echocardiography.
I have some comments mainly referred to the methodology which will hopefully further improve the quality of the manuscript.
Authors’ response: we thank the Reviewer for the time and effort spent in revising our manuscript and for all useful comments and suggestions, which we addressed point-by-point below.
Can you please provide further information regarding the variable selection of your multivariable regression model.
Authors’ response: we thank the Reviewer for this question. Covariates were tested in the multivariable regression model if they were positive in univariable analysis. We have now added this information to the revised manuscript in the statistical methods section.
Can you provide some information regarding the fact that you used an alpha of 0.005 in your sample size estimation.
Authors’ response: we apologize for the typing error and thank the Reviewer for having recognized it. The alpha level was set at 0.05.
We have now revised the sentence in the revised manuscript accordingly.
But the most important issue in my eyes is the great amount of tested hypotheses. You should definitely implement adjustment for multiplicity in your analysis, in order to draw firm conclusions.
Authors’ response: we thank the Reviewer for this suggestion. We fully agree with the Reviewer that the multiple tests performed should be taken into account in the analysis.
Correction for multiple comparisons were applied to statistical analyses. Specifically, as GWI, GCW, GWW and GWE present are not completely independent, adjustment with the Holm method was applied in this case. To account for comparison in multiple independent subgroups (gender groups, LF-LG) adjustment with the Bonferroni method was applied.
Adjustment of our analyses for multiple comparisons confirmed our results in al cases.
In addition, myocardial work indices were repeatedly measured at different time points. Thus, despite the main comparison was between pre-TAVI and post-TAVI (2 weeks) values, we applied a repeated measure fixed-effect model to account for time-correlation over time. Using this model, myocardial work indices were used as the within-subject factor, and 4 time-points were considered (pre-TAVI, 2 weeks, 4 months, 1 year). Using the repeated measures computation model we confirmed statistical significance for GWI and GCW but not for GWE. However, this has no relevant impact on the clinical relevance of our results, since the numerical difference underlying the unadjusted statistical significance previously found for GWE at 1 year was very modest to have clinical relevance.
In fact, this significance wasn’t even highlighted in the original version of the manuscript:
“On the contrary, we observed no significant change in GWW nor GWE (Figure 2).”
These details have been added to the revised manuscript and the absence of statistical significance for GWE at 1 year has been clearly noted on the manuscript.
Round 2
Reviewer 2 Report
The authors adequately addressed all topics mentioned during the review process.